# Miniaturized Bandpass Filter Using a Combination of T–Shaped Folded SIR Short Loaded Stubs

**DOI:** 10.3390/s22072708

**Published:** 2022-04-01

**Authors:** Kicheol Yoon, Kwang Gi Kim, Tae-Hyeon Lee

**Affiliations:** 1Medical Devices R&D Center, Gachon University Gil Medical Center, 21, 774 Beon-gil, Namdong-daero, Namdong-gu, Incheon 21565, Korea; kcyoon98@gachon.ac.kr; 2Department of Biomedical Engineering, College of Medicine, Gachon University, 38-13, 3 Beon-gil, Dokjom-ro 3, Namdong-gu, Incheon 21565, Korea; 3Department of Biomedical Engineering, College of Health Science, Gachon University, 191 Hambak-moero, Yeonsu-gu, Incheon 21936, Korea; 4Department of Health Sciences and Technology, Gachon Advanced Institute for Health Sciences and Technology (GAIHST), Gachon University, 38-13, 3 Beon-gil, Dokjom-ro, Namdong-gu, Incheon 21565, Korea; 5Department of Electronic Engineering, Gyeonggi University of Science and Technology, 269 Gyeonggigwagi-daero, Siheung 15073, Korea; thlee@gtec.ac.kr

**Keywords:** T–shaped SIR, J–inverter, bandpass filter, *λ_g_*/4 transmission line, stub

## Abstract

The consumption of multimedia content is ubiquitous in modern society. This is made possible by wireless local area networks (W–LAN) or wire service systems. Bandpass filters (BPF) have become very popular as they solve certain data transmission limitations allowing users to obtain reliable access to their multimedia content. The BPFs with quarter–wavelength short stubs can achieve performance; however, these BPFs are bulky. In this article, we propose a compact BPF with a T–shaped stepped impedance resonator (SIR) transmission line and a folded SIR structure. The proposed BPF uses a T–shaped SIR connected to a J–inverter structure (transmission line); this T–shaped SIR structure is used to replace the *λ_g_*/4 transmission line seen in conventional stub BPFs. In addition, a folded SIR is added to the short stubs seen in conventional stub BPFs. This approach allows us to significantly reduce the size of the BPF. The advantage of a BPF is its very small size, low insertion loss, and wide bandwidth. The overall size of the new BPF is 2.44 mm × 1.49 mm (0.068*λ_g_* × 0.059*λ_g_*). The proposed BPF can be mass produced using semiconductors due to its planar structure. This design has the potential to be widely used in various areas including military, medical, and industrial systems.

## 1. Introduction

In the era of the fourth industrial revolution, the internet of things (IoT) based on big data and artificial intelligence (AI) applications have become explosively popular [1]. The reason is that in the fourth industrial revolution, artificial intelligence learning technology is often used, and artificial intelligence learning technology operates an automation system through big data collection and processing. At this time, most of the data for big data collection has a high capacity, and broadband communication is utilized for large–capacity processing. Moreover, to increase portability, the size of the system is becoming smaller. Bandpass filters (BPFs) are an essential component in the microwave devices that are usually used in both RF front–ends and as transmission components in fourth industrial revolution systems [2]. In addition, the fourth industrial revolution, big data, autonomous driving functions, and sensors are increasingly used. In the case of a bandpass filter, it is sometimes applied as a sensor that detects a resonance phenomenon and passes a desired frequency band [3,4]. The sensor–based bandpass filter is also applied to medical systems for diagnosis or treatment [3,4]. Among BPFs, stub BPFs are known for having a simple structure and being easy to produce. In addition, a stub BPF’s bandwidth can be adjusted (narrow or wide) through the tapped line technique [5,6,7]. This has led to stub BPFs being in great demand.

Moreover, quarter–wavelength short stubs BPFs are often used instead of quarter–wavelength open stub BPFs when the vertical size of the BPF must be reduced [8,9]. On the other hand, there is not a design that can reduce the horizontal–size of the transmission line (TL) in BPFs with *λ_g_*/4 short or *λ_g_*/2 open stubs (18.7 mm × 18.9 mm @ 5.8 GHz and 0.45*λ_g_* × 0.19*λ_g_*) [5,10]. Currently, in order to reduce the size of a BPF, a high dielectric (*ε_r_*) substrate is used, and the cost of this approach is very high. As such, there is great interest in designing BPFs with desirable characteristic in terms of a good response, small size, and low cost [11].

There have been numerous studies to reduce the size of stub BPFs while ensuring a wide bandwidth [12,13,14]. The authors of [12] used the zeroth–order resonance (ZOR) method producing a 11.9 mm × 6.0 mm BPF with a bandwidth of 119%. However, this BPF used a coupling structure for the feeding line, which generated coupling losses. The BPF proposed in [13] was 19.88 mm × 22.62 mm in size and had a bandwidth of 116%; the BPF used a coupling structure for its in/output part and employed the DGS (defected ground structure) technique. The disadvantages of this BPF, in addition to being quite bulky, were the coupling losses it generated and the parasitic phenomenon between the transmission line and ground that occurred. In [14], a BPF was introduced that had a wide bandwidth of 109% but was also quite large in size (23.4 mm × 10 mm). In addition, this BPF used the DGS technique. Its parasitic performance between the transmission line and ground was bad, while its insertion losses were high. Studies [15,16,17] described BPFs that were 32 mm × 10 mm, 13.0 mm × 10.0 mm, and 150 mm × 150 mm, respectively, in size and had wide bandwidths (123% [15,16] and 146% [17]). However, these devices also had high insertion losses (0.8 dB [13], 0.3 dB [14], and 0.55 dB [17]).

In this article, we present a compact BPF with *λ_g_*/4 short stubs that uses a T–shaped SIR transmission line and a folded SIR structure. The T–shaped SIR transmission line and folded SIR are designed to further reduce horizontal–TL and short stubs as well as the vertical length. By replacing the series *λ_g_*/4 TL and shunt *λ_g_*/4 short stubs, which protrude along the respective horizontal and vertical axes, with T–shaped and folded SIRs, the proposed BPF achieves a more compact size. The compact a BPF has a simple structure and low insertion losses.

## 2. Analysis for Phenomenon of a BPF

A conventional BPF with *λ_g_*/4 short stubs coupled to a *λ_g_*/4 transmission line (TL) and *λ_g_*/4 short stubs is shown in Figure 1 [5,9]. From the figure, *Z_T_* is the characteristic impedance of a *λ_g_*/4 transmission line, and the *Z_s_* and *Z_sm_* are characteristic of *λ_g_*/4 short stubs, respectively. They have electrical lengths of 90 degrees. Then, the electrical parameters of a conventional BPF is provided in Table 1, in which we report the *Z_s_*, *Z_sm_* and *Z_t_* respectively. The *θ_s_*, *θ_sm_* and *θ_T_* are the electrical lengths of the short stubs and transmission line (TL) in the conventional BPF. A transmission line is operated with the admittance impedance (J–inverter).

Figure 2 shows the simulation result of the Smith chart. In the figure, the real and imaginary impedance are 0.001 + j0.082 with a bandwidth of 120% at a center frequency of 2.45 GHz in the conventional BPF. However, the proposed BPF does not have the structure and conditions of a conventional BPF.

The proposed BPF is composed of a T–shaped SIR transmission line and a folded short stub, as shown in Figure 3a. In the figure, *Z_s_* and *Z_sm_* are the characteristic impedance of the folded short stubs, while *Z_T_* is the characteristic impedance of the T–shaped transmission line. As shown in the figure, *θ_s_* and *θ_sm_* are the electrical length of the folded short stubs, while *θ_T_* is the electrical length of the T–shaped transmission line.

Figure 3b shows the equivalent circuit of the proposed BPF. The equivalent circuit of a proposed BPF consists of a resonant circuit (*C_s_*//*L_s_* and *C_sm_*//*L_sm_*) and J–inverter (*J*_0,1_ and _0,2_); the resonant circuit is connected to the inductors (*L_s_* and *L_sm_*) and capacitors (*C_s_* and *C_sm_*) in the short–circuited shunt stubs with *λ_g_*/4, as shown in Figure 3c. The J–inverter consists of a *λ_g_*/4 transmission line [5,9]. *L_s_* and *C_s_* are the respective inductance and capacitance corresponding to the short stub *Z_s_,* while the *L_sm_* and *C_sm_* are the inductance and capacitance corresponding to the stub (*Z_sm_*). In Figure 3c, the J–inverter is configured with the *λ_g_*/4 transmission line (TL), which has a phase response of 90 degrees. In addition, the shunted short stubs are configured with the inductance (*L_s_* and *L_sm_*) and capacitance (*C_s_* and *C_sm_*) resonant circuits. Then, the short stubs with the inductance and capacitance resonant circuits have a phase response of 90 degrees. Here, the *Y_o_* is the characteristic admittance, and the *Y_s_* and *Y_sm_* are characteristic of admittance for short stubs. *Y_T_* is characteristic of the admittance of the transmission line. Figure 4 shows the SIR structure; the *Z_s_* and *Z_sm_* are both high impedance SIR, while *Z_T_* is a low impedance SIR, the input impedance (*Z_i_*) values are given by Equations (1) and (2) [18,19], where *Y_i_* is the input admittance.
(1)$$Z_{i}=jZ_{sm}\frac{Z_{s,sm}tan\;\theta_{s,sm}-Z_{T}tan\;\theta_{T}}{Z_{T}-Z_{s,sm}tan\;\theta_{T}tan\;\theta_{s,sm}}=0$$
(2)$$Y_{i}=\underset{\theta_{i}\to 90{}^{\circ}}{\lim}\frac{j\left(2tan\;\theta_{i}-\frac{Z_{i2}}{Z_{i1}tan\;\theta_{i}}\right)}{Z_{T}+\frac{Z_{T}^{2}}{Z_{s,sm}}-Z_{s,sm}tan^{2}\;\theta_{s,sm}}=0$$

The SIR can be obtained from the response of the electrical length corresponding to the short–loaded stub type, which is shown in Equation (3).
(3)$$\;\theta_{i}=arctan\left(\sqrt{\frac{Z_{T}}{2Z_{s,sm}}}\right)=\pi -arctan\left(\sqrt{\frac{Z_{T}}{2Z_{s,sm}}}\right)=\frac{\pi }{2}$$

When a resonant phenomenon occurs, such as those shown in Figure 3a and Figure 4, *Y_i_* becomes zero (*Y_i_* = 0). Then, at the resonant frequency, the electrical length and impedance matching must be zero (*θ_i_* = 90°) at *Z_i_* = *Z_T_*/Z*_s_*_,*sm*_. where *Z_i_* is the input impedance. The calculated impedance and electrical length for the equivalent circuit are given in Table 2. Then, the real and imaginary impedance of the proposed BPF are 0.001 + j0.083 with a phase response of 90 degrees at 2.45 GHz, as shown in Figure 5, which is similar to Figure 2 (0.001 + j0.082) with a phase response of 90 degrees at 2.45 GHz.

Figure 6 shows the simulation results for the equivalent circuit of a BPF. From the figure, the filter shows the simulated frequency is 2.45 GHz (0.94 to 3.88 GHz) with a bandwidth of 120%, and the insertion and return losses are 0.09 dB and 24.8 dB, respectively.

## 3. Design and Fabrication

The structure of a proposed BPF has a T–shaped SIR transmission line and folded SIR short stubs whose size can be reduced as shown in Figure 7a. This figure shows how the folded SIR structure (*Z_s_* and *Z_sm_*), which has high impedance (short circuit loaded), and the T–shaped SIR structure (*Z_T_*), which has low impedance (middle term), are applied to the short stubs and transmission line (J–inverter).

Figure 7b shows the design of the proposed BPF. Here, it is shown that *l_sa_*, *l_sb_*, and *l_sc_* are 0.51 mm, 0.19 mm and 0.14 mm, respectively, while *l_sm_*, *l_T_*, and *l_v_* are 1.17 mm, 0.99 mm and 0.16 mm, respectively. w_s_, w_sm_, w_T_ and w_v_ are 0.16 mm, 0.14 mm, 0.32 mm and 0.20 mm, respectively, while *l* × *h* are 2.44 × 1.49 mm^2^. In addition, the via hole has a diameter of Ø0.3 mm. The *l_v_* structure gives extra space between the folded SIR short stub and feeding line. If the lv was not between them, the folded SIR short stubs and feeding line could become connected, and there would be a danger of a short circuit. As such, the physical length of *l_v_* should be within *λ_g_*/8.

Figure 7c shows the proposed BPF fabricated on a Teflon substrate that has a low dielectric constant of 2.54 and a height of 0.54 mm. In the fabrication process, the film is developed through development, water, fixer, and drying processes through X–ray imaging, as shown in Figure 7d, to manufacture the filter. The substrate is put on the spin quarter, the photo resistor (TPR) solution is applied, and the spin operation is performed. Then, after UV–patterning of the substrate and film, the TPR solution is sacrificed using TPR development. Finally, the production of the filter is completed through wet etching [20].

## 4. Experimental Results

The results from the simulations and experiments on the proposed BPF with a T–shaped SIR transmission line and folded SIR short stubs are shown in Figure 8. As seen in the figure, the filter through simulation gave insertion and return losses of 0.10 dB and 24.9 dB, respectively, and a bandwidth of 120% at a center frequency of 2.45 GHz (0.94 to 3.88 GHz band), the experiment gave insertion and return losses of 0.10 dB and 19.2 dB, respectively, with a bandwidth of 120% at a center frequency of 2.44 GHz (0.85 to 3.85 GHz band).

Table 3 compares the proposed BPF with others. The comparison includes details on the devices’ bandwidth, insertion loss, and total size.

In the size comparison process, [21,22,23,24,25,26] had higher frequency and permittivity values than the proposed filter. Nevertheless, the size of the proposed filter was analyzed to be smaller than [21,22,23,24,25,26]. Therefore, the small size of the proposed filter is very meaningful.

The center frequency and fractional bandwidth of the designed BPF were 2.45 GHz and 120%, respectively, which was applied to a 4G long–term evolution (LTE) mobile system. The operated frequency and bandwidth of a 4G LTE mobile system is 4 GHz (2 to 8 GHz) and 120%. However, the frequency of the proposed BPF is 2.45 GHz because the 2.45 GHz is an unlicensed frequency band. The designed filter offers suggestions for size and bandwidth tunability. Therefore, the filter used unlicensed frequencies [27].

## 5. Conclusions

We proposed a compact bandpass filter (BPF) with a T–shaped stepped impedance resonator (SIR) transmission line and folded SIR short stubs. The proposed BPF adopted a T–shaped SIR transmission line and folded SIR short stubs instead of the *λ_g_*/4 transmission line and quarter–wavelength short stubs found in conventional stub BPFs. The advantages of this BPF are its small size, low insertion losses, and wide bandwidth.

As conventional stub BPFs are relatively large, the filter design’s compact size makes it an important step forward for BPFs. The BPF applies the SIR technique, which makes it possible to reduce the device’s size while maintaining excellent performance. The BPF can be increased in fractional bandwidth. The overall size of the proposed design is 2.44 × 1.49 mm^2^ (0.068*λ_g_* × 0.059*λ_g_*), a significant improvement over the current design. In addition, the proposed BPF can easily be mass produced thorough the use of semiconductors due to its planar structure. The design of a filter has a potential to be applied in a variety of situations including military, medical, and industrial systems.

## Figures and Tables

**Figure 1 sensors-22-02708-f001:**
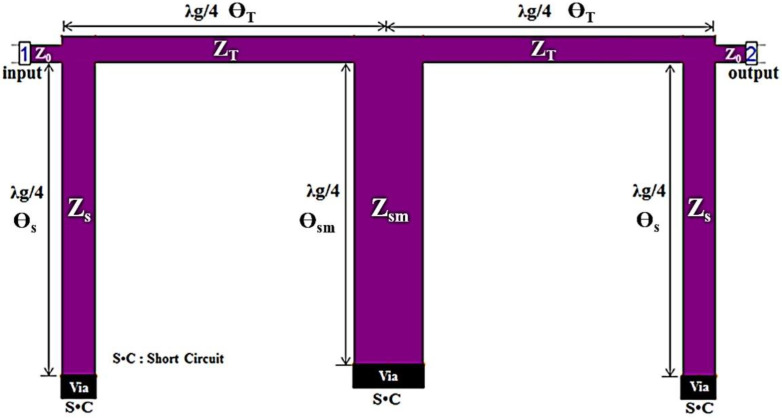
Conventional BPF with *λ_g_*/4 short stubs.

**Figure 2 sensors-22-02708-f002:**
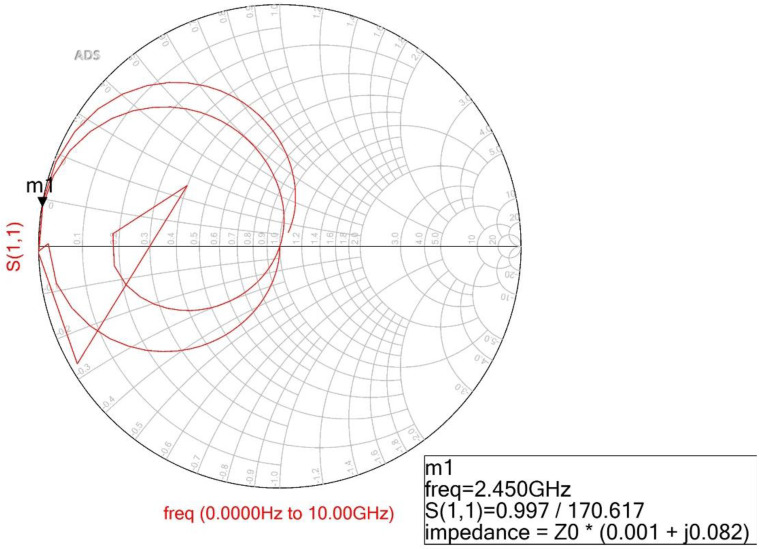
Simulation result for real and imaginary impedances for a conventional BPF.

**Figure 3 sensors-22-02708-f003:**
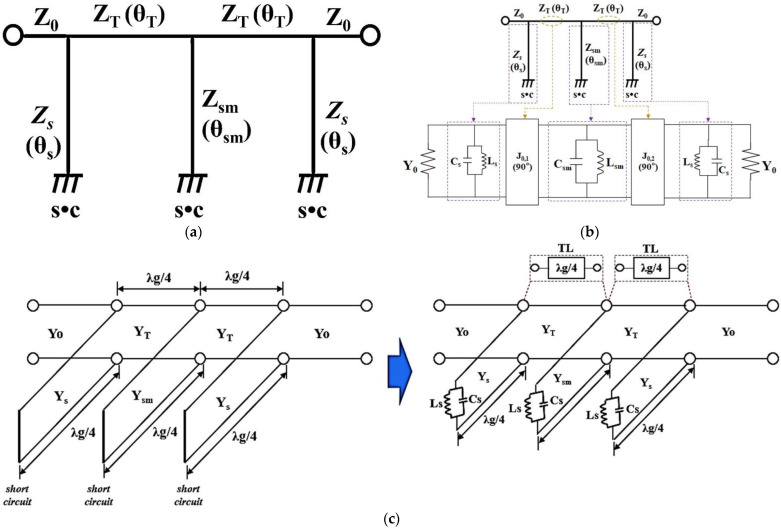
Equivalent circuit of a proposed BPF. (**a**) BPF. (**b**) J–inverter. (**c**) Detailed equivalent circuit.

**Figure 4 sensors-22-02708-f004:**
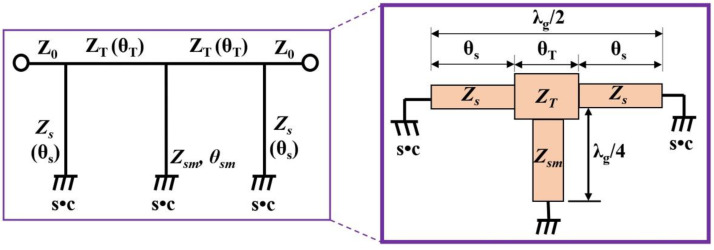
SIR structure of the proposed BPF with transmission line and short stub.

**Figure 5 sensors-22-02708-f005:**
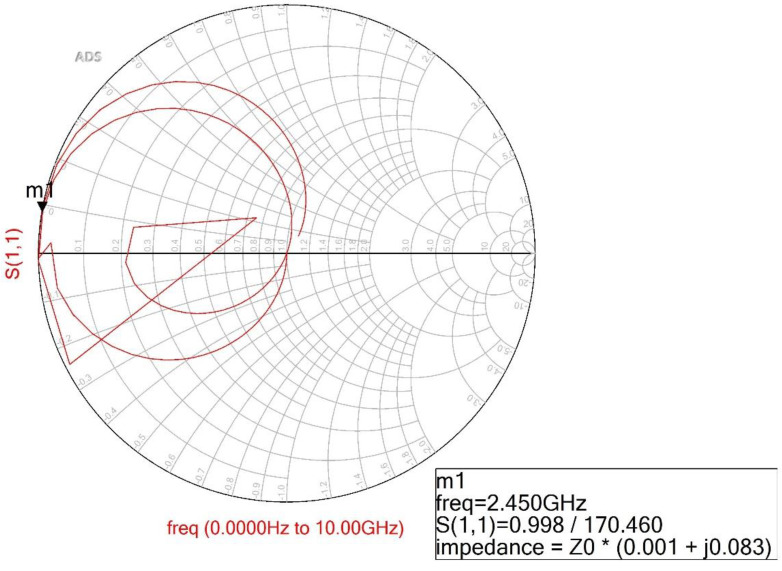
Simulation result for real and imaginary impedances for the proposed BPF.

**Figure 6 sensors-22-02708-f006:**
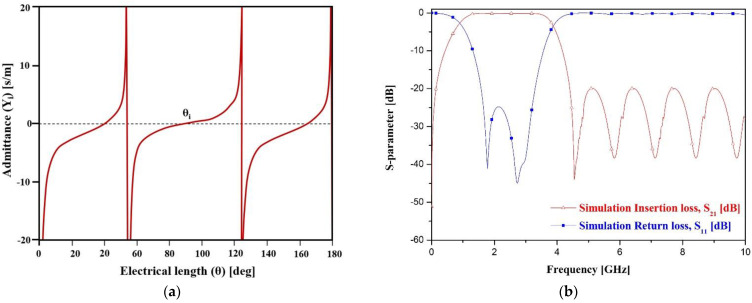
Simulation results for electrical response (**a**) and electrical length by admittance (**b**) bandwidth.

**Figure 7 sensors-22-02708-f007:**
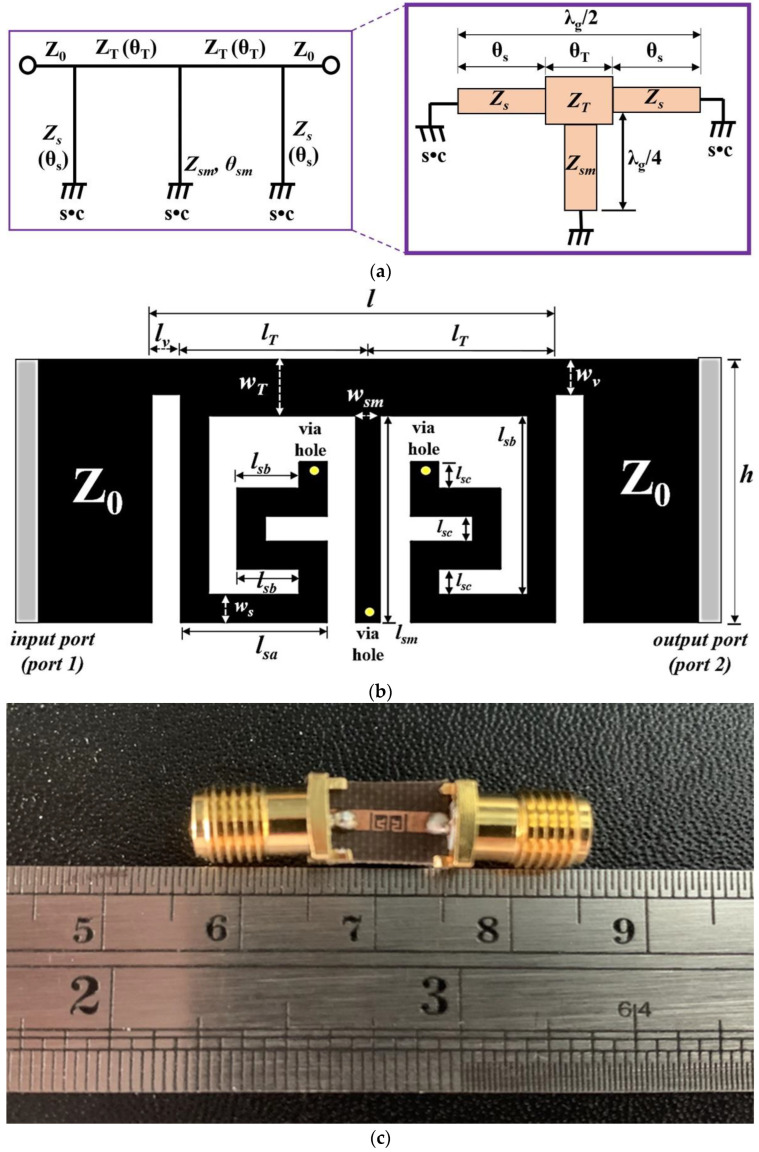
Structure of the proposed BPF: (**a**) SIR structure; (**b**) layout; (**c**) fabrication; and (**d**) fabrication process (wet etching).

**Figure 8 sensors-22-02708-f008:**
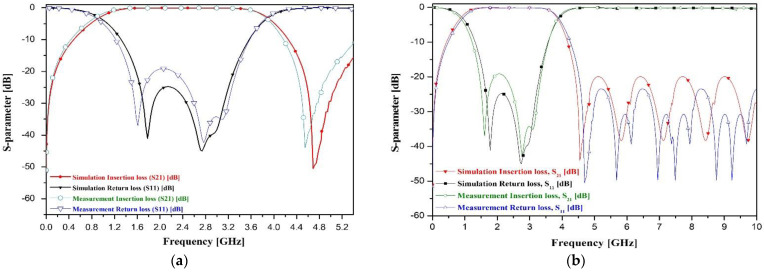
Experimental results for the proposed BPF: (**a**) narrow scale frequency band and (**b**) wide scale frequency band.

**Table 1 sensors-22-02708-t001:** Calculated electrical parameters of conventional BPF.

Parameter	Value [Ω]	Parameter	Value [deg]
*Z_s_*	256.1	*θ_s_*	90.0
*Z_sm_*	148.0	*θ_sm_*	90.0
*Z_T_*	37.30	*θ_T_*	90.0

**Table 2 sensors-22-02708-t002:** Calculated electrical parameters for the proposed BPF.

Parameter	Value [Ω]	Parameter	Value [deg]
*Z_s_*	89.2	*θ_s_*	2.31
*Z_sm_*	142	*θ_sm_*	2.79
*Z_T_*	75.0	*θ_T_*	2.82

**Table 3 sensors-22-02708-t003:** Comparison between the proposed BPF and others.

Ref [#]	Center Frequency [GHz]	IL [dB]	RL [dB]	BW [%]	Size [λg]	Dielectric Constant
This work	2.44	0.10	19.2	120	0.068 × 0.059	2.45
[21]	2.45	0.10	15.0	59	0.171 × 0.136	3.30
[22]	6.75	0.80	11.0	100	0.60 × 0.54	3.55
[23]	4.83	1.10	10.5	131	0.29 × 0.275	2.20
[24]	7.20	1.45	17.0	111	0.69 × 0.18	2.20
[25]	6.85	1.50	15.0	109	1.10 × 0.40	3.55
[26]	6.95	0.42	19.0	97	0.128 × 0.37	3.55

## Data Availability

The data presented in this study are available upon request from the corresponding author. The data are not publicly available because of privacy and ethical restrictions.

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
