# Peer review of "Miniaturized Bandpass Filter Using a Combination of T–Shaped Folded SIR Short Loaded Stubs"

_sensors, 2022, doi:10.3390/s22072708_

Round 1

Reviewer 1 Report

The authors propose a miniaturized band-pass filter with inductive coupling method using combination of T-shaped folded SIR short loaded stub.

The authors describe the design, the method, the simulated and measured results. This is an interesting paper, however I have some remarks and questions about the paper such as:

  • The English prose has to be improved.
  • What are Zi and Yi linked to the equation (1) and (2)? Could the authors add some details? The synthesis has to be detailed.
  • In line number 108, the simulated results are noted as “the insertion and return losses are 0.09 dB and 24.8 dB” whereas line 136 it is written “the insertion and return losses are 0.1 dB and 24.9 dB”… That is to be corrected.
  • In line number 127, it is noted that “the via hole has a diameter of 3Ø”, the unit has to be added and “and a height of 0.576 mm” whereas the substrate height is 0.54mm, whereas the photograph do not permit to see the via hole, I do not know if that is a mistake.
  • In the title, the Authors say that the filter use an inductive coupling method, but that only appear in the title.

Author Response

Point 1: The English prose has to be improved.

Response 1: I am currently undergoing proofreading by an English expert. I will finish English proofreading before publication..

Point 2: What are Zi and Yi linked to the equation (1) and (2)? Could the authors add some details? The synthesis has to be detailed.

Response 2: Zi and Yi are input impedance. I added the means of Zi and Yi.

Please refer to word lines of 125-126 (green).

Point 3: In line number 108, the simulated results are noted as “the insertion and return losses are 0.09 dB and 24.8 dB” whereas line 136 it is written “the insertion and return losses are 0.1 dB and 24.9 dB”… That is to be corrected.

Response 3: Thanks for the good point 0.1dB is a typo, 0.09dB is correct. So I corrected the insertion loss.

Please refer to word lines of 149 (sky blue).

Point 4: In line number 127, it is noted that “the via hole has a diameter of 3Ø”, the unit has to be added and “and a height of 0.576 mm” whereas the substrate height is 0.54mm, whereas the photograph do not permit to see the via hole, I do not know if that is a mistake.

Response 3: I'm not sure why I added this value (0.576mm). I think this value is my mistake. So 0.576mm was deleted. Thanks for pointing it out.

Please refer to word lines of 170 (yellow).

Point 5: In the title, the Authors say that the filter use an inductive coupling method, but that only appear in the title.

Response 5: It has nothing to do with the method of inductive coupling in the title. thanks for the good point In the title, the inductive coupling method was deleted. So I edited the title.

Reviewer 2 Report

This manuscript presents the design and characterization of a reduced size BPF with a T-shaped stepped impedance resonator (SIR) transmission-line and a folded SIR structure.

The antenna design and the measurements presented represent a great merit of this work.

However, the manuscript itself does not have the characteristics of a research. The document, as it is presented, particularly in sections 2, 3 and 4, looks like a report.

Not enough information about the design process, measurements, manufacture (plannar CMOS?), and testbench is available to allow replication by other researchers.

The englisha lso needs to be revised. Some passages are difficult to understand, especially in sections 1 and 2.

Author Response

Point 1: The antenna design and the measurements presented represent a great merit of this work. However, the manuscript itself does not have the characteristics of a research. The document, as it is presented, particularly in sections 2, 3 and 4, looks like a report.

Response 1: This paper is not an antenna design. This paper is a bandpass filter with reduced size and increased bandwidth. Please note. I identified the lack of content and supplemented the analysis and sentences for the completeness of the thesis.

Please refer to word lines of 84-91, 96-99, 111-114, 117-125, and 138-140 (gray).

Figure 1, Figure 2, Figure 3, Figure 5, and Table 1.

Point 2: Not enough information about the design process, measurements, manufacture (plannar CMOS?), and testbench is available to allow replication by other researchers.

Response 2: I made an addition to Figure 3(d). I also added a sentence on lines 174-180. Please check the yellow text.

Point 3: The englisha lso needs to be revised. Some passages are difficult to understand, especially in sections 1 and 2.

Response 3: I am currently undergoing proofreading by an English expert. I will finish English proofreading before publication.

Reviewer 3 Report

Dear authors, thank you for this BPF-paper, it is instructive. Please find my comments below:

The introductional part needs to be reworked, it is not very precise.

Line 36: "speed modernization has continually incresed" - I'm not sure I know what that means at all, to me it is an empty phrase

Line 39: Put a point after AI, and rephrase the following phrase. maybe "In todays' data-driven society, wide-band communication os more vital then ever."

Line 54: howcome you are talking about lambdag/4 and lambdag/2 stubs and all of a sudden you reference 0.45 lambdag and 0.19 lambdag ?

Line 63: Explain DGS

Lines 58 to 72: Does all these filters work at the same frequency? Otherwise size comparison is not meaningful.

In general: Howcome you give no intended specification for your filter? The design is made at 2.45 GHz for Wi-Fi? What is the channel bandwidth that you need than, how are specified the rejection and stopbands? What is the intended usage of your filter?

Line 89: "the resonant circuit is connected to the inductors (L s  and L sm ) and capacitors (C s  and C sm )"... to me  the resonant circuits consist of the inductors and capacitors. Please explain

Figure2: Simulation is done from 0-4.5 GHz, you could go higer in frequency to show if there is a second pass band or not.

Line 127: The via hole diameter is how much? 3Ø ? I'm not sure I know how much that is? What is the unit?

Figure4: Experimental measurements stop at 5.2 GHz, but we see a climbing of the insertion loss graph, I'd like to know what happens higher up in frequency as the tendency is rising?

You didn't explain if you only fabricated one sample or several.  The difference between simulated and measured results, is it because of tolerances in dimensions? A retrosimulation could help.

Table2: ref[22] size is presented in mm and not in lambdag as the rest.

Thank you, and have a nice day.

Author Response

Point 1: Dear authors, thank you for this BPF-paper, it is instructive. Please find my comments below: The introductional part needs to be reworked, it is not very precise.

Response 1: In order to improve the quality of the paper, your detailed comments are very positive. I appreciate your comments.

Point 2: Line 36: "speed modernization has continually incresed" - I'm not sure I know what that means at all, to me it is an empty phrase Line 39: Put a point after AI, and rephrase the following phrase. maybe "In todays' data-driven society, wide-band communication os more vital then ever."

Response 2: The sentence on line 39 has been supplemented. Please refer to the yellow sentences on lines 36-42 of the introduction.

Point 3: Line 54: howcome you are talking about lambdag/4 and lambdag/2 stubs and all of a sudden you reference 0.45 lambdag and 0.19 lambdag ?.

Response 3: The stub BPFs of λg/4 and λg/2 have already been studied for a long time. This filter has undergone a lot of research to reduce it to a practical size because of the size issue. So, the λg/4 and λg/2 stub BPFs under the 5.8GHz frequency condition are approximately 0.45 λg and 0.19 λg in length of the stub. Therefore, I add references [5], [10] to prove this size. Reference is next to the figure. However, the reference has been moved to the end of the sentence. However, this paper is 2.45 GHz. The frequency of this paper is lower than that of [5] and [10]. Nevertheless, the size of this paper is smaller than the frequencies of [5] and [10]. So I added references and figures in the hopes of giving it a smaller size.

Point 4: Line 63: Explain DGS.

Response 4: I put the full name (word lines 66-67, yellow).

Point 5: Lines 58 to 72: Does all these filters work at the same frequency? Otherwise size comparison is not meaningful.

Response 5: Yes. All right! Not the same frequency. However, the size of the filter in this paper is smaller than [21]-[26]. The reason is that [21]-[26] have higher frequency and permittivity values than the proposed filter. Nevertheless, the size of the filter is larger than the size of the proposed filter.

Please refer to word lines of 198-201 (green).

Point 6: In general: Howcome you give no intended specification for your filter? The design is made at 2.45 GHz for Wi-Fi? What is the channel bandwidth that you need than, how are specified the rejection and stopbands? What is the intended usage of your filter?

Response 6: Please refer to word lines of 202-208 (yellow).

The center frequency and fractional bandwidth of designed a BPF are 2.45 GHz and 120%, respectively which is applied to 4G LTE () mobile system. The operated frequency and bandwidth of a 4G LTE (Long Term Evolution) mobile system is 4 GHz (2 to 8 GHz) and 120%. However, the frequency of the proposed BPF is 2.45 GHz. The reason for using 2.45 GHz is that 2.45 GHz is an unlicensed frequency band. The designed filter makes suggestions for size and bandwidth tunability. Therefore, the filter used unlicensed frequencies [27].

Point 7: Line 89: "the resonant circuit is connected to the inductors (L s  and L sm ) and capacitors (C s  and C sm )"... to me  the resonant circuits consist of the inductors and capacitors. Please explain Figure2:

Response 7: Figure 3 and word lines 111-148 were corrected and supplemented. Thanks for pointing it out.

In particular, it was analyzed that the calculated impedance (real and imaginary) of the proposed filter was almost identical to that of the conventional filter through smith-chart simulations (Figures 2 and 5).

Point 8: Simulation is done from 0-4.5 GHz, you could go higer in frequency to show if there is a second pass band or not.

Response 8: After setting the frequency band from 0 GHz to 10 GHz, experiments (simulation and measurement) were performed again. At 5.2 GHz, it is a cut-off band.

Please refer to Figure 6 and Figure 8.

Point 9: Line 127: The via hole diameter is how much? 3Ø ? I'm not sure I know how much that is? What is the unit?

Response 9: Revised to Ø0.3mm.

Please refer to word lines of 169 (yellow).

Point 10 : Figure4: Experimental measurements stop at 5.2 GHz, but we see a climbing of the insertion loss graph, I'd like to know what happens higher up in frequency as the tendency is rising?

You didn't explain if you only fabricated one sample or several.

Response 10: After setting the frequency band from 0 GHz to 10 GHz, experiments (simulation and measurement) were performed again. At 5.2 GHz, it is a cut-off band.

Please refer to Figure 6 and Figure 8.

Point 11: The difference between simulated and measured results, is it because of tolerances in dimensions? A retrosimulation could help.

Response 11: Differences between simulation results and measurement results are very common. The environment analyzed by the simulation tool and the real environment are very different. The actual environment is inevitably different due to errors in the manufacturing process and the load resistance component of the measuring equipment. Therefore, in the field of RF-filter design, it is generally acceptable for the error range of simulation and measurement results to reach within 5%.

Point 12: Table2: ref[22] size is presented in mm and not in lambdag as the rest.

Response 12: 23.0 mm x 20.0 was revised to 0.29λgx0.275λg (see Table 3).

Round 2

Reviewer 1 Report

The suggestions and corrections were taken into account.

Author Response

Point 1: The suggestions and corrections were taken into account.

Response 1: You pointed out carefully to improve the completeness of my paper.

Therefore, I would like to thank you very much for your paper review.

Reviewer 2 Report

Sorry about "The antenna design". When I was reviewing your work I started thinking about how to use your process in the fractal antenna project I've been working on. So, by my mistake, escaped the term antenna.

Author Response

Point 1: Sorry about "The antenna design". When I was reviewing your work I started thinking about how to use your process in the fractal antenna project I've been working on. So, by my mistake, escaped the term antenna.

Response 1: The filter and antenna follow the resonator theory, so thank you very much for your good point. Thanks to this, I believe my paper has improved.

Reviewer 3 Report

Dear authors, your paper is much improved, you have responded to the different comments in a nice way.

To me, I would skip lines 36-41 in your introduction and start with Line 42.

Some english faults are still there (lines 42, 76, 90, 98, 117, 119, maybe others) , thus the article should be revised to improve the language.

Otherwise, nice job!

Best regards

Author Response

Point 1: Dear authors, your paper is much improved, you have responded to the different comments in a nice way. To me, I would skip lines 36-41 in your introduction and start with Line 42. Some English faults are still there (lines 42, 76, 90, 98, 117, 119, maybe others) , thus the article should be revised to improve the language.

Otherwise, nice job!

Response 1: Your point was a great help to improve my paper. Thank you very much for your good evaluation. Also, I will review the English sentences as a whole and continue to revise and supplement them before publishing.
